# Discussion of Design Wind Loads on a Vaulted Free Roof

**Wei Ding and Yasushi Uematsu \***

National Institute of Technology (KOSEN), Akita College, Akita 011-8511, Japan; tei85@akita-nct.ac.jp
* Correspondence: uematsu@akita-nct.ac.jp; Tel.: +81-18-847-6001

**Abstract:** This paper discusses the wind loads for designing vaulted free roofs based on a wind tunnel experiment, in which the wind force coefficients for the main wind force resisting system and the peak wind force coefficients for cladding are considered. The focus is on the dynamic load effects of fluctuating wind pressures on the wind force coefficients. Wind pressure distributions on the top and bottom surfaces were measured in a turbulent boundary layer. The results indicated that the distributions of wind force coefficients changed significantly with wind direction. Then, the wind direction providing the maximum load effect on the structural frame was detected from a dynamic response analysis using the time histories of wind pressure coefficients. In the analysis, the focus was on the bending moment at the windward column base and the axial force in the leeward column as the most important load effects. The LRC method proposed by Kasperski was employed for evaluating the equivalent static wind force coefficients providing the maximum load effects. Based on the results, a model of design wind force coefficient was proposed in the framework of the conventional gust effect factor approach. Finally, positive and negative peak wind force coefficients for designing the cladding were proposed based on the most critical maximum and minimum peak wind force coefficients among all wind directions.

**Keywords:** wind load; vaulted free roof; main wind force resisting system; cladding; wind tunnel experiment; LRC method; conditional sampling; gust effect factor

## 1. Introduction

Free-standing canopy roofs, or free roofs providing shade and weather protection are often constructed in public spaces, such as parks, shopping centers and sports grounds, all over the world (see Figure 1). The roofs are supported by only columns in most cases. Because both the top and bottom surfaces are exposed to turbulent winds, the net wind forces acting on the roof become very complicated compared with usual buildings. The roofs are generally sensitive to dynamic wind forces because of their lightness and flexibility. Therefore, the wind resistant performance is one of the most important issues in the structural design of these roofs.

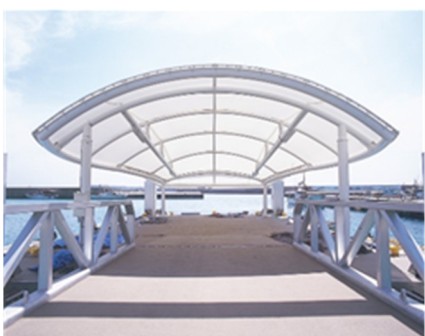

**Figure 1.** A vaulted free roof (provided by Taiyo Kogyo Corporation).

A large number of experimental and/or numerical studies have been made regarding the wind pressures on enclosed buildings of various configurations, the results of which have been incorporated into building codes and standards. By comparison, the number of studies of wind loads on free roofs is quite limited. This is probably due to difficulties in making models as well as in measuring wind pressures in wind tunnel experiments. We should install many pressure taps on both the top and bottom surfaces of the roof for measuring the distributions of net pressures on the roof in detail. However, it is difficult to install so many pressure taps because the roof thickness and column width should be small so that they would not affect the wind field around the roof and the wind pressure distributions on the roof significantly.

Many researches have been conducted on the wind loads of planar free roofs, i.e., gable, troughed and mono-sloped roofs (e.g., Gumley [1], Letchford and Ginger [2], Ginger and Letchford [3], Natalini et al. [4] and Uematsu et al. [5,6]). These researches have provided basic data for the specification of wind force coefficients (net wind pressure coefficients) in building codes and standards (e.g., ASCE 7 [7] and AIJ Recommendations for Loads on Buildings [8]). On the other hand, the number of studies of wind loads on curved free roofs is rather limited. This is probably due to a further difficulty in making thin curved models. The Australian/New Zealand standard [9] provides the net wind pressure coefficients for hyperbolic paraboloid (HP)-shaped free roofs with a limited range of rise/span ratio. Recently, wind loads and wind-induced responses of HP-shaped free roofs have been investigated by several researchers. Kaseya et al. [10] made a wind tunnel measurement of wind force coefficients on an HP-shaped free roof. Nakagawa et al. [11] numerically analyzed the dynamic responses of an HP-shaped membrane roof using the time histories of wind force coefficients obtained from a wind tunnel experiment. The thickness of their wind tunnel models was as thick as 5 mm, whereas the width was only 100 mm. Such a thick model may deform the flow around the roof and the resultant wind pressure distributions on the roof significantly [1,4]. Colliers et al. [12] established a hybrid rapid prototyping methodology for making double curved canopy structures. They successfully made thin models and measured the wind pressures on such models. However, the measurement was limited to the mean pressure coefficients. Uematsu et al. [13] discussed the design wind loads on an HP-shaped free roof based on a wind tunnel experiment, in which the overall aerodynamic forces and moments on rigid roof models were measured by a six-component force balance. They proposed a model of wind force coefficients for the main wind force resisting systems considering the dynamic load effects of fluctuating wind pressures. The wind tunnel models were made of nylon resin by using laser lithography, which made the roof thickness as small as 2 mm. Recently, Uematsu and Yamamura [14] carried out a wind tunnel experiment of the wind pressure distributions on domed free roofs with rise/span ratios ranging from 0.1 to 0.4. A 3D printer was used for making the wind tunnel models. The roof thickness was only 2 mm, whereas the diameter was 150 mm. Many pressure taps were installed along a centerline both on the top and bottom surfaces. The pressure distribution on the whole roof was measured by rotating the model.

Natalini et al. [15] measured the wind forces on vaulted free roofs in a turbulent boundary layer. The rise/span ratio was 0.2, and the side ratio (depth/span ratio) was 2 or 4. The measurement was limited to the mean wind pressures on the roof. The dynamic load effect of fluctuating wind pressures was not considered. Uematsu and Yamamura [16] measured the overall aerodynamic forces and moments acting on vaulted free roofs using a six-component force balance. The side ratio of the models was fixed to 1, but the rise/span ratio was changed from 0.1 to 0.4. They measured the distributions of net pressures along two representative arc lines; one was along the centerline and the other was along the verge. They used a 3D printer to make the wind tunnel models. The roof thickness was 1 mm for the overall wind force measurement models and 2 mm for the wind pressure measurement models. Wen et al. [17] made a similar pressure measurement using the same models. They changed the model's side ratio from 1 to 3 using one or two dummy models without pressure taps. They measured the distributions of wind pressure coefficients only along

several arc lines. Then, they made a CFD simulation using Large Eddy Simulation (LES). They regenerated the flow around the roof almost successfully. However, the computations were made for limited cases due to a long computation time required; the rise/span ratio was 0.1 or 0.4 and the wind direction was 90° (normal to the eaves) or 45° (diagonal).

The present study discusses the wind loads of a vaulted free roof with a side ratio of 1 and a rise/span ratio of 0.1 based on a wind tunnel experiment. The model used here is similar to that of Uematsu and Yamamura [16], but it has many pressure taps. Twenty pressure taps are distributed over a half area of the roof. We assume that the roof is supported by two rigid frames conmprising two columns and a beam, each of which carries the wind loads acting on the corresponding half area. We focus on the bending moment at the windward column base and the axial force in the leeward column as the load effects when discussing the design wind loads. The wind direction that generates the largest load effect is first detected from a time-history response analysis of the frame. Next, we obtain the distribution of equivalent static wind pressure coefficients providing the largest load effect is obtained using the LRC method proposed by Kasperski [18]. Then, we propose a simple model of design wind force coefficients in the framework of the conventional gust effect factor approach. The distributions of the maximum and minimum peak wind force coefficients over the whole roof area are also measured. Based on the results for the most critical maximum and minimum peak wind force coefficients among all wind directions, we propose the positive and negative peak wind force coefficients for the design of cladding. Although the tested case is limited a case where the side ratio is 1 and the rise/span ratio is 0.1, this study will provide a useful reference for the wind-resistant design of this type of free roof.

## 2. Wind Tunnel Model

The present study considers a vaulted free roof, as shown in Figure 2a, which has a square plan of 16.7 m ($B$) × 16.7 m ($L$). The rise/span ratio ($f/B$) is 0.1, which is a representative value for practical structures. Note that a survey of practical structures constructed in Japan indicated that the values of $f/B$ were generally in a range from 0.1 to 0.2 and many of them were around 0.1. The mean roof height $H$ is 8 m, which is the same as that of our previous studies [16,17]. We assume that the roof is supported by two steel rigid frames as shown in Figure 2b and that the wind load on each half of the roof (Area 1 or 2) is supported by each frame (Frame 1 or 2). Because the frame is a statically indeterminate structure, it is necessary to determine the sections of the members for computing the stresses induced in the members (load effects). The columns (square steel tubes) and the beam (H-section steel) are rigidly jointed, and the column base is fixed to the ground. The dimensions of the members were determined by using a conventional allowable stress design method. The design wind loads were provided by the mean wind loads for a design wind speed $U_H$ (= 28.6 m/s) at the mean roof height $H$ multiplied by a gust effect factor $G_f$. The value of $U_H$ was calculated following the procedure specified in the Building Standard Law of Japan assuming that the 'reference wind speed' $V_0$ was 36 m/s. The mean wind load was provided by the product of the velocity pressure $q_H$ (= $\frac{1}{2}\rho U_H^2$, where $\rho$ is air density) and the mean wind force coefficient $\overline{C}_f$, obtained from the wind tunnel experiment. The gust effect factor $G_f$ is also specified in the Building Standard Law of Japan, which is 2.5 for Terrain Category III (suburban exposure). The dimensions of the members determined were as follows:

Column: 300 mm (width) × 300 mm (depth) × 12 mm (thickness) (JIS G 3466)

Beam: 300 mm (height) × 200 mm (width) × 8 mm (web thickness) × 12 mm (flange thickness) (JIS G 3192)

where 'JIS' means 'Japan Industrial Standard'. The design strength of the material, corresponding to the yield stress, is 235 N/mm$^2$.

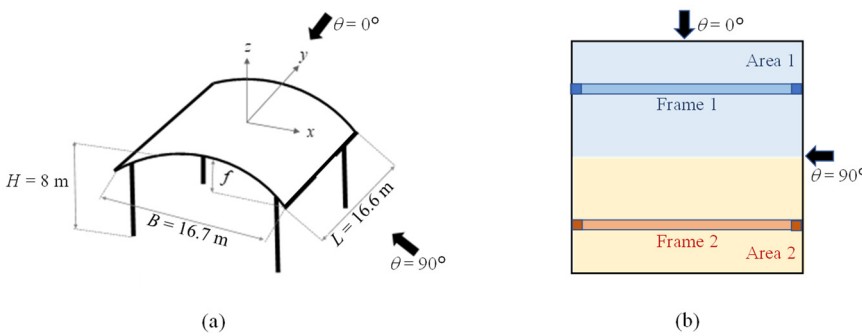

(a)                                                               (b)

**Figure 2.** Investigated building: (**a**) General view; (**b**) structural system.

Figure 3a shows the wind tunnel model used in this study. The geometric scale is $\lambda_L = 1/100$. The model has six square columns of 7 mm width. Note that these columns do not reproduce the practical columns. The roof comprises a sandwich structure with a thickness of 4 mm. Each of the top and bottom surfaces has twenty pressure taps (0.6 mm ID) arranged in a half area (Area 1), as shown in Figure 3b. Each pressure tap is connected to a pressure transducer (Wind Engineering Institute, MAPS-02) using a tubing that consists of a copper pipe (0.6 mm ID) and a flexible vinyl tube (1 mm ID); the total length of the tubing is 1 m. Six columns are necessary to lead forty copper pipes to the underside of the wind tunnel. It is expected that the columns affect the flow around the roof and the resultant pressure distribution on the roof only slightly, because the width is 7 mm, as will be shown in the next section.

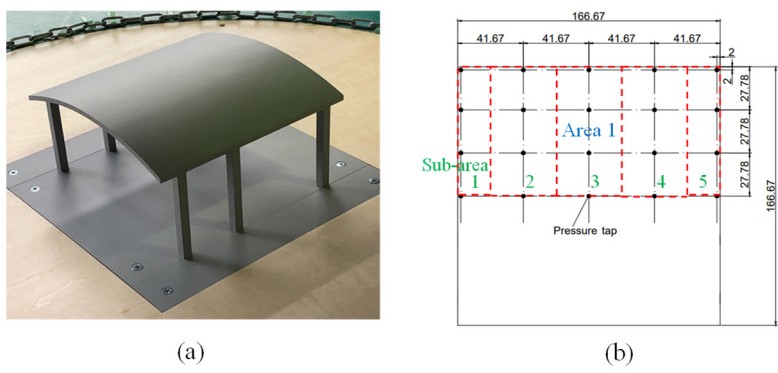

(a)                                                               (b)

**Figure 3.** Wind tunnel model: (**a**) Picture; (**b**) pressure tap arrangement.

## 3. Experimental Procedure

### 3.1. Wind Tunnel Flow

The pressure measurements are conducted in a wind tunnel of Eiffel type at the Department of Architecture and Building Science, Tohoku University, Japan, which has a working section of 1.4 m width, 1.0 m height and 6.5 m length. The wind tunnel flow is a turbulent boundary layer. The profile of mean wind speed $U_z$ is approximated by a power law with an exponent of $\alpha = 0.27$ (see Figure 4a, where $U_z$ is normalized by the value at a reference height, $z_{ref} = 600$ mm; and $z$ represents the height from the wind tunnel floor). The turbulence intensity $I_{uH}$ at the mean roof height $H$ is about 0.2. The non-dimensional power spectrum, $fS_u(f)/\sigma_u^2$, of wind speed fluctuation at a height of $z = 10$ cm (nearly equal to $H$) is shown in Figure 4b, where $S_u(f)$ = power spectrum, $f$ = frequency, $\sigma_u$ = standard deviation of fluctuating wind speed, and $L_x$ = integral length scale of turbulence. The general shape of the spectrum is consistent with the Karman-type spectrum with $L_x \approx 0.2$ m. The value of $L_x$ is about one third of the specified value in the AIJ Recommendations for Loads on Buildings [8], which is 0.58 m (= 58 m/100). However, such a disagreement is acceptable for low-rise buildings judging from the Tieleman et al.'s criterion [19–21] that the $L_x$ value of the wind tunnel flow at the roof height should be

larger than both 0.2 times the target value (= 0.2 × 0.58 = 0.12 m) and twice the roof height (= 2 × 0.09 = 0.18 m).

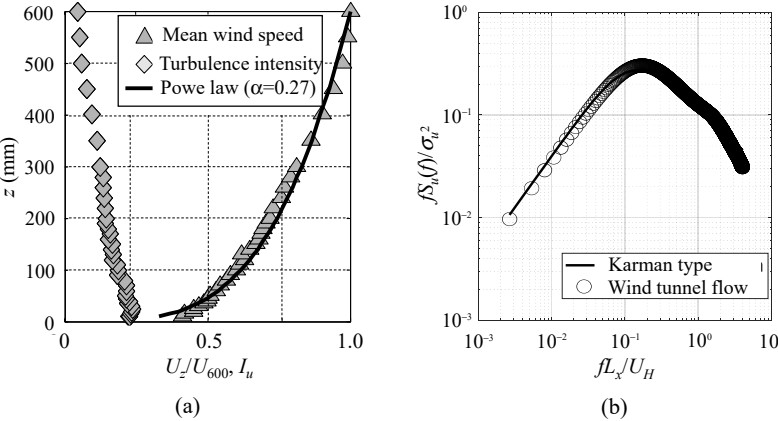

**Figure 4.** Characteristics of wind tunnel flow: (**a**) Profiles of mean wind speed $U_z$ and turbulence intensity $I_u$; (**b**) non-dimensional power spectrum of fluctuating wind speed at a height of z = 10 cm.

The mean wind speed $U_H$ at a height of $H$ is set to 9 m/s in the wind tunnel experiment. Because the design wind speed is $U_H$ = 28.6 m/s, as mentioned above, the velocity scale of wind tunnel flow is calculated as $\lambda_V$ = 1/3.18, which results in the time scale of $\lambda_T$ (= $\lambda_L/\lambda_V$) = 1/31.4. The Reynolds number $Re$, defined by $Re = U_H H/\nu$, where $\nu$ represents the kinematic viscosity of air, is about $4.8 \times 10^4$. It is well known that $Re$ affects the flow around a curved structure significantly. This feature is related to a shift of flow separation point on the curved surface with $Re$. In the case of $f/B$ = 0.1, the flow over the roof's top surface does not separate from the roof surface when $\theta \approx 90°$, as will be shown later. Furthermore, the flow separation occurs at the windward edges (eaves and verge) in a diagonal wind, which gives the largest load effects, as will be described later. The results at a diagonal wind direction will be used for proposing the design wind force coefficients. These features imply that the present results can be applied to the practical design of vaulted free roofs with $f/B$ = 0.1. Detailed discussion of the Reynolds number effect on the flow around vaulted free roofs is presented in Uematsu and Yamamura [16].

The Reynolds number $Re$ and the blockage ratio $Br$ of the model satisfy the requirements of the ASCE Wind Tunnel Testing for Buildings and Other Structures [22]; i.e., $Br$ < 5% and $Re > 1.1 \times 10^4$.

*3.2. Pressure Measurement*

Pressures at forty taps are sampled simultaneously at a sampling frequency of 500 Hz for a sampling time of 19.1 s which is equivalent to 10 min at full scale. Considering the variation of data, the measurement is repeated 18 times. The statistical values of pressures, such as the maximum and minimum pressures, are evaluated by applying ensemble averaging to the results of 18 runs. A low-pass filter with a cut-off frequency of 300 Hz is used for eliminating high frequency noise. The distortion in amplitude and phase of measured pressures due to tubing is corrected by using an appropriate transfer function of the tubing system. The wind direction $\theta$, defined as shown in Figure 2, is varied from 0° to 180° at an increment of 15° considering the symmetry of the model. The pressure distributions in Area 2 for $\theta$ = 0°–90° can be obtained from those in Area 1 for $\theta$ = 90°–180°.

The measured pressure $P$ is reduced to a non-dimensional pressure coefficient $C_p$ as

$$C_P = \frac{P - P_s}{q_H} \tag{1}$$

where $P_s$ represents the static pressure, which is measured at a height of $z_{ref}$ above the model's center. The wind pressure coefficients on the top and bottom surfaces of the model

are represented by $C_{pt}$ and $C_{pb}$, respectively. The net wind force per unit area (i.e., pressure difference) acting on the roof is given by the difference between the pressures on the top and bottom surfaces, which is normalized by $q_H$. As a result, the wind force coefficient $C_f$ may be given by the following equation:

$$C_f = C_{pt} - C_{pb} \qquad (2)$$

Therefore, the sign of $C_f$ is the same as that of $C_{pt}$.

## 4. Wind Force Coefficients for the Main Wind Force Resisting System

### 4.1. Mean Wind Pressure and Force Coefficients

Figures 5–8 show the distributions of the mean wind pressure coefficients, $\overline{C}_{pt}$ and $\overline{C}_{pb}$, on the top and bottom surfaces and the mean wind force coefficients, $\overline{C}_f$, at typical wind directions, i.e., $\theta = 0°, 45°, 60°$ and $90°$. Any distinct effects of columns on the $\overline{C}_{pb}$ and $\overline{C}_f$ distributions can be seen in the figures. This implies that the effect of columns on the wind forces on the roof is relatively small. When $\theta = 0°$ (Figure 5), large suctions occur near the windward edge (verge). This feature may be related to the flow separation at the roof's leading edge accompanied with the flow reattachment on the roof surface. Because the roof thickness is 4 mm, the suction area is very small. The other area is subjected to small suctions; the values of $\overline{C}_{pt}$ and $\overline{C}_{pb}$ are approximately $-0.2$. The values of $\overline{C}_f$ are nearly equal to zero over the whole area, because the pressures on the top and bottom surfaces cancel out each other. The distributions of $\overline{C}_{pt}$, $\overline{C}_{pb}$ and $\overline{C}_f$ show a little asymmetry with respect to the centerline parallel to the wind direction, which may be due to unavoidable experimental errors. In diagonal winds, such as $\theta = 45°$ (Figure 6) and $60°$ (Figure 7), large suctions are induced near the verge. Such high suctions may be caused by a conical vortex generated over the roof along the verge. As a result, large magnitude negative wind force coefficients occur in this area. The area near the windward edge is subjected to positive pressures on the top surface due to impinging winds, while negative pressures on the bottom surface due to flow separation. The combination of these positive and negative pressures causes large positive wind force coefficients in this area. When $\theta = 90°$ (Figure 8), large suctions are induced on the top surface in the central area. Because the rise/span ratio $(f/B)$ of the roof is 0.1, the flow separation from the top surface does not occur [16,17]. The flow along the roof is accelerated in the direction of wind, and the wind speed becomes the maximum near the roof top, which generates the minimum pressure coefficient. On the other hand, the flow separates downward at the windward edge (eaves) of the roof, inducing large suctions on the bottom surface near the windward edge. As a result, large positive wind forces are generated in the windward area, while large magnitude negative wind forces are generated in the central area. It is interesting to note that the contour lines are almost perpendicular to the wind direction, which means that the time-averaged pressure on the roof is approximately two dimensional even when $L/B = 1$.

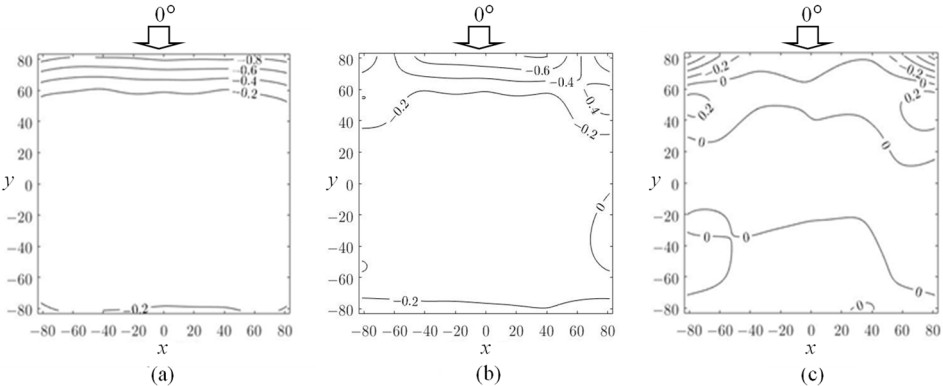

**Figure 5.** Contours of mean wind pressure and force coefficients at $\theta = 0°$: (**a**) $\overline{C}_{pt}$, (**b**) $\overline{C}_{pb}$, (**c**) $\overline{C}_f$.

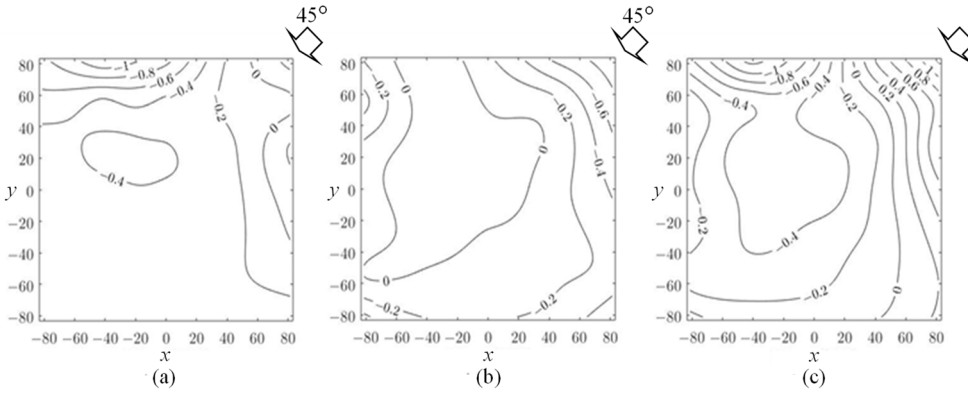

**Figure 6.** Contours of mean wind pressure and force coefficients at θ = 45°: (**a**) $\overline{C}_{pt}$, (**b**) $\overline{C}_{pb}$, (**c**) $\overline{C}_f$.

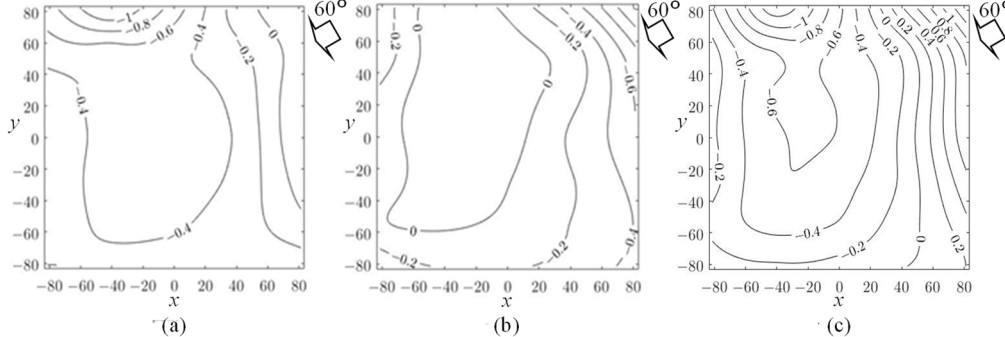

**Figure 7.** Contours of mean wind pressure and force coefficients at θ = 60°: (**a**) $\overline{C}_{pt}$, (**b**) $\overline{C}_{pb}$, (**c**) $\overline{C}_f$.

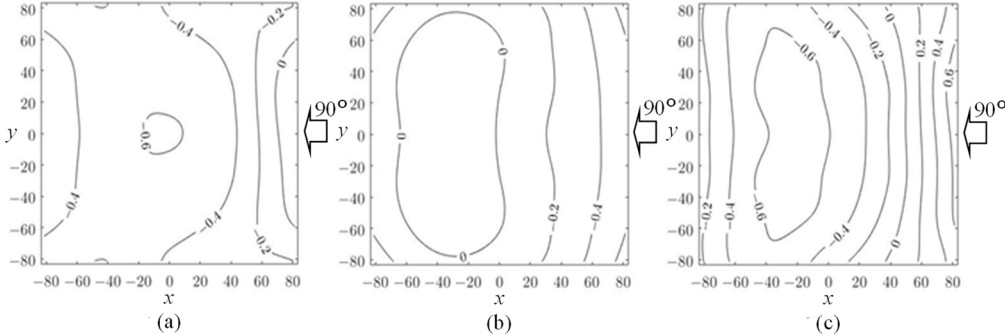

**Figure 8.** Contours of mean wind pressure and force coefficients at θ = 90°: (**a**) $\overline{C}_{pt}$, (**b**) $\overline{C}_{pb}$, (**c**) $\overline{C}_f$.

*4.2. Estimation of the Equivalent Static Wind Force Coefficients*

The wind loads for designing the main wind force resisting system of the free roof, represented by equivalent static wind loads, are evaluated base on the Load Resistance Correlation (LRC) method that Kasperski [18] proposed. The validity of this method is confirmed by the results of a conditional sampling method, as will be shown later. It is assumed that the structure is so rigid that the resonance effect of fluctuating wind pressures on the structural responses can be neglected. When using this method, we should recognize the load effects to be considered for evaluating the design wind loads. Regarding small-scale rigid structures, Uematsu et al. [23] mentioned that such load effects could be selected from a static response analysis of the structure subjected to the time-averaged wind loads. Following this procedure, we found that the most important load effects are the bending moment $M(t)$ at the windward column base and the axial force $N(t)$ induced in the leeward column. These two load effects dominate the structural design of the vaulted free roof

under consideration. Note that $N(t)$ is related to the pull-out force acting on the foundation. $M(t)$ and $N(t)$ are provided by the following equations:

$$M(t) = q_H \sum_{j=1}^{5} \alpha_j C_{f,j}(t) A_j \tag{3}$$

$$N(t) = q_H \sum_{j=1}^{5} \beta_j C_{f,j}(t) A_j \tag{4}$$

where $\alpha_j$ and $\beta_j$ are the influence coefficients for $M(t)$ and $N(t)$, respectively, that is, the values of the bending moment and the axial force when a concentrated load of 1 N is applied to a point '$j$' on the frame ($j$ = 1–5). The point of application corresponds to the location of the pressure-tap line parallel to the eaves (see Figure 3b). $C_{f,j}(t)$ and $A_j$, respectively, represent the area-averaged wind force coefficients and the tributary area for each sub-area of the roof (see Figure 3b). $C_{fj}(t)$ is provided by a weighted average of wind force coefficients $C_{f,jk}$ at four pressure taps ($k$ = 1–4) located in each sub-area; the weight is proportional to the tributary area of each pressure tap. Then, $M(t)$ and $N(t)$ are normalized as follows:

$$M^*(t) = \frac{M(t)}{q_H B^2 (L/2)} \tag{5}$$

$$N^*(t) = \frac{N(t)}{q_H B (L/2)} \tag{6}$$

The maximum peak values of $M^*(t)$ and $N^*(t)$, denoted by $\hat{M}^*$ and $\hat{N}^*$, respectively, during a sampling time of 600 s at full scale are directly obtained from the time histories of $M^*(t)$ and $N^*(t)$. Figure 9a,b, respectively, show the variation of $\hat{M}^*$ and $\hat{N}^*$ with wind direction $\theta$. As mentioned above, the pressure measurement was repeated 18 times at each wind direction. Therefore, we can obtain 18 data for each of $\hat{M}^*$ and $\hat{N}^*$. The plotted value at each wind direction is the ensemble average of the 18 data. It is found that the largest values of $\hat{M}^*$ and $\hat{N}^*$, denoted by $\hat{M}^*_{cr}$ and $\hat{N}^*_{cr}$, respectively, occur at an oblique wind direction $\theta_{cr}$ ($\approx 60°$) not at a normal wind direction ($\theta$ = 90°). This is probably due to high suctions caused by a conical vortex generated over the roof along the verge (see Figure 7). Hereafter, focus is on $\theta$ = 60° for discussing the design wind force coefficients.

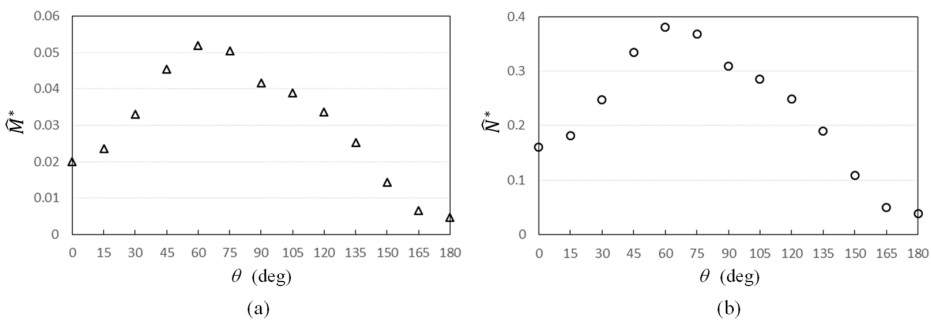

**Figure 9.** Variation of $\hat{M}^*$ and $\hat{N}^*$ with wind direction θ: (**a**) $\hat{M}^*$; (**b**) $\hat{N}^*$.

Using the LRC method, we can obtain the equivalent static wind force coefficients, denoted as '$C_{f\_LRC}$' hereafter, providing the maximum load effect as follows:

$$C_{f\_LRC} = \overline{C}_f + g_r C'_f \rho_{rf} \tag{7}$$

where $g_r$ is a peak factor of the load effect; $C'_f$ is the standard deviation of fluctuating wind force coefficient; and $\rho_{rf}$ is the coefficient of correlation between wind force and response (load effect). These values can be obtained from the time histories of wind force coefficient and load effects. The practical value of $g_r$ was found to be 4.1 for both $M^*(t)$

and $N^*(t)$, which is somewhat larger than the value (=3.5) that Kasperski [18] assumed. This is probably because the probability density function of the load effect deviates from the Gaussian distribution.

In addition to the LRC method, a conditional sampling method and a gust effect factor approach are employed to discuss the design wind force coefficients in more detail. The conditional sampling method provides the distribution of wind force coefficients, denoted as '$C_{f\_cond}$' hereafter, at a moment when the maximum load effect ($\hat{M}^*$ or $\hat{N}^*$) occurs. In the gust effect factor approach, the distribution of wind force coefficients, denoted as '$C_{f\_gust}$' hereafter, is provided by the product of $\overline{C}_f$ and a gust effect factor, $G_f$. $G_f$ is defined by the ratio of the maximum peak value to the mean value of the load effect. The value of $G_f$ was found to be approximately 2.2 for $\hat{M}^*$ and 2.1 for $\hat{N}^*$. Based on the quasi-steady assumption, $G_f$ is approximated by the following equation:

$$G_f \approx G_v^2 \approx (1 + g_v I_{uH})^2 \tag{8}$$

where $G_v$ and $g_v$ represent a gust factor and a peak factor of the approaching flow, respectively. Substituting $G_f \approx 2.2$ or $2.1$ and $I_{uH} \approx 0.2$ into Equation (8), we obtain $g_v \approx 2.3$ or $2.4$. These values of $g_v$ are somewhat smaller than that for planar free roofs, which is approximately 3.0 [6]. This difference in $g_v$ may be due to a small difference in the characteristics of the flow separation. That is, the turbulence generated by the flow separation is lower for vaulted roofs than for planar roofs, resulting in a smaller value of $g_v$. Such smaller values of $g_v$ were observed for other curved free roofs, i.e., HP-shaped and domed free roofs [13,14]. Using Equation (8), we can obtain the value of $G_f$ for any other turbulent flow using the provided $I_{uH}$ value.

The distributions of the equivalent static wind force coefficients predicted by the above-mentioned three methods are shown in Figure 10; focus is on $\hat{M}^*$ and $\hat{N}^*$ in Figure 10a,b, respectively. In the figures, the distribution of the mean wind force coefficients, represented by $C_{f\_mean}$, is also presented for reference. The abscissa of the figure is the distance $s$ from the windward edge of the roof along the circular arc, normalized by its maximum value, $s_{max}$. It is found that the distributions of $C_{f\_cond}$ is consistent with those of $C_{f\_LRC}$. This seems natural judging from the principle of the LRC method. The difference between $C_{f\_LRC}$ and $C_{f\_cond}$ may be due to the limited number of samples. It is interesting that the $C_{f\_gust}$ distribution is also similar to the $C_{f\_LRC}$ distribution, although the magnitude of $C_{f\_gust}$ is somewhat larger than that of $C_{f\_LRC}$. Comparing the values of load effects predicted by the $C_{f\_gust}$ and $C_{f\_LRC}$ distributions with each other, we found that the ratio was 1.25 for the bending moment $M$ and 1.34 for the axial force $N$. That is, the gust effect factor approach overestimates the load effects by about 30%. If such an overestimation is acceptable, the gust effect factor approach can be used effectively, because it is a very simple procedure requiring only the time-averaged wind force coefficients.

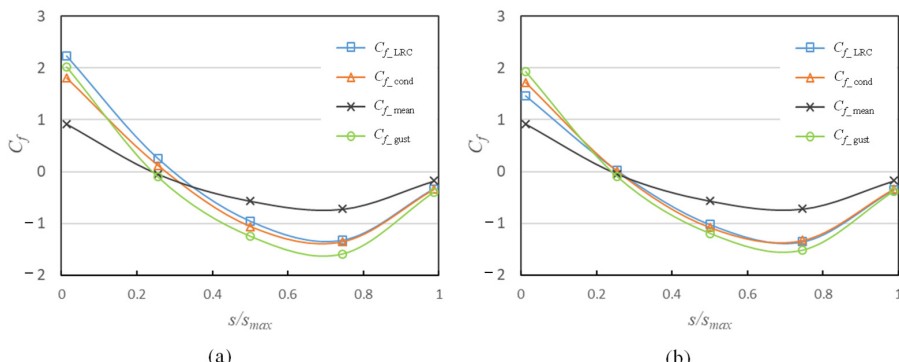

(a) (b)

**Figure 10.** Distributions of equivalent static wind force coefficients providing the maximum load effects: (**a**) Bending moment $M$; (**b**) axial force $N$.

We can use the $C_{f\_LRC}$ distribution as it is in the structural design. However, a simple model of wind force coefficient may be more useful in practical design. The simplest model may be as follows. That is, the roof is divided into three zones, $R_U$, $R_M$ and $R_L$, and constant values of $C_f$, denoted by $C_{f\_U}$, $C_{f\_M}$ and $C_{f\_L}$, are provided to these three areas, as shown in Figure 11. The zoning is the same as that for cylindrical roofs of enclosed buildings in the AIJ Recommendations for Loads on Buildings [8]. The values of $C_{f\_U}$, $C_{f\_M}$ and $C_{f\_L}$ are provided, respectively, by integrating $C_f$ over the three zones, $R_a$, $R_b$ and $R_c$. The area-averaged values of these wind force coefficients are shown in Table 1. Then, the effect of area-evraging of $C_{f\_LRC}$ on the load effects ($M$ and $N$) is examined by comparing the predicted values of $M$ and $N$ from the $C_{f\_U}$, $C_{f\_M}$ and $C_{f\_L}$ values in Table 1 and those from the practical $C_{f\_LRC}$ distributions shown in Figure 10a,b. It is found that the ratio is 1.08 for $M$ and 1.04 for $N$. That is, a model of $C_f$ distribution represented by $C_{f\_U}$, $C_{f\_M}$ and $C_{f\_L}$ overestimates the load effects just a little bit. In other words, this model can be used for practical design reasonably.

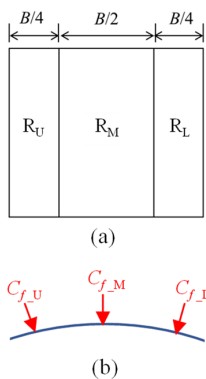

**Figure 11.** Zoning of vaulted free roof: (**a**) Plan; (**b**) section.

**Table 1.** Area-averaged equivalent wind force coefficients.

| Load Effect Considered | $C_{f\_U}$ | $C_{f\_M}$ | $C_{f\_L}$ |
|---|---|---|---|
| Axial force $N$ | 0.74 | −1.11 | −0.85 |
| Bending moment $M$ | 1.23 | −1.05 | −0.83 |

In many building codes and standards, the wind loads for designing the main wind force resisting systems are usually provided by the product of wind force coefficients and a gust effect factor. Following this approach, the design wind force coefficients, repesented by $C^*_{f\_U}$, $C^*_{f\_M}$ and $C^*_{f\_L}$, are given by the $C_{f\_U}$, $C_{f\_M}$ and $C_{f\_L}$ values divided by the above-mentioned gust effect factor, $G_f$ (2.2 for $M$ and 2.1 for $N$). The proposed wind force coefficients are listed in Table 2.

**Table 2.** Proposed wind force coefficients for the amin wind force resisting systems.

| Load Effect Considered | | $C^*_{f\_M}$ | $C^*_{f\_L}$ |
|---|---|---|---|
| Axial force $N$ | 0.35 | −0.53 | −0.41 |
| Bending moment $M$ | 0.56 | −0.48 | −0.38 |

## 5. Peak Wind Force Coefficients for Designing Cladding

Several researchers have investigated the methods for estimating the wind loads for designing cladding of buildings considering various factors, e.g., Cook and Mayne [24,25], Harris [26,27], Kasperski [28,29], and Hui et al. [30]. In the AIJ Recommendations for Loads on Buildings [8], however, the design wind load for cladding is provided by the product of the velocity pressure $q_H$, peak wind force coefficient $\hat{C}_f$ and subjected area $A_c$ of cladding, and positive and negative peak wind force coefficients, $C_{f\_pos}$ and $C_{f\_neg}$, are specified.

The specified values of $C_{f\_pos}$ and $C_{f\_neg}$ are determined based on the mean values of the maximum and minimum peak wind force coefficients considering the variations of the peak values. Following such a procedure, this paper focuses on the ensemble averages of the 18 data for the maximum and minimum peak wind force coefficients obtained in the wind tunnel experiment.

Because the subjected area of cladding $A_c$ is generally small, the values of $C_{f\_pos}$ and are usually specified based on the most critical maximum and minimum peak values of area-averaged wind force coefficients over the subjected area irrespective of wind direction [8]. The values of $C_{f\_pos}$ and $C_{f\_neg}$ generally decrease in magnitude with an increase in $A_c$. In this study, however, because the number of pressure taps is rather limited, we cannot discuss the effect of subjected area on $C_{f\_pos}$ and $C_{f\_neg}$ in detail. Therefore, we focus on the most critical maximum and minimum wind force coefficients, $\hat{C}_{f,\mathrm{cr}}$ and $\check{C}_{f,\mathrm{cr}}$, among all wind directions, which were obtained at each point.

According to Uematsu and Isyumov [31] who investigated the relationship between time and spatial averages for peak wind pressures on the roof and wall corners of a gable-roofed low-rise building under the condition that the pressure field was approximately regarded homogeneous from the statistical viewpoint, the effect of time average over a period of $T_a$ on the minimum peak pressure coefficient is approximately equivalent to that of spatial average over an area given by $A = \left(T_a U_H / k_p\right)^2$, where $k_p$ represents a decay constant of fluctuating pressures. It is thought that this is the case for the fluctuating wind forces acting on the vaulted free roof under consideration. Then, we assume that $k_p = 4.5$ [32,33] and $A = 1$ m$^2$ as representative values for $k_p$ and $A$, respectively [8]. Substituting these values together with $U_H = 28.6$ m/s into the above equation, we obtain $T_a = 0.16$ s. Aplying a moving average of $T_a = 0.16$ s to the time histories of $C_f$, we can obtain the maximum and minimum peak values of $C_f$, represented by $\hat{C}_f$ and $\check{C}_f$, directly from the smoothed time histories of wind force coefficients.

Figures 12–15 show the contours of the maximum and minimum peak wind force coefficients, $\hat{C}_f$ and $\check{C}_f$, at some typical wind directions, i.e., $\theta = 0°$, $45°$, $60°$ and $90°$. Note that any distinct effect of columns on the peak wind force coefficients is not observed. When $\theta = 0°$, the wind forces fluctuate similarly both in the positive and negative directions. The magnitude of $\hat{C}_f$ and $\check{C}_f$ is almost the same, and the maximum value of about 1.2 occurs only in the vicinity of the windward edge (verge). In the other areas the magnitude is about 0.4. In diagonal winds, e.g., when $\theta = 45°$ and $60°$, very large values of $\hat{C}_f$ occur near the windward edge, whereas the magnitude of $\check{C}_f$ becomes very large near the windward verge. The areas subjected to large positive and negative peak wind forces correspond well to those subjected to large mean wind forces (see Figures 6 and 7). A similar feature can be observed for $\theta = 90°$. That is, large values of $\hat{C}_f$ occur near the windward edge (eaves), whereas the magnitude of $\check{C}_f$ becomes large near the roof top. However, the magnitude of $\hat{C}_f$ and $\check{C}_f$ is not so large compared with that observed in diagonal winds.

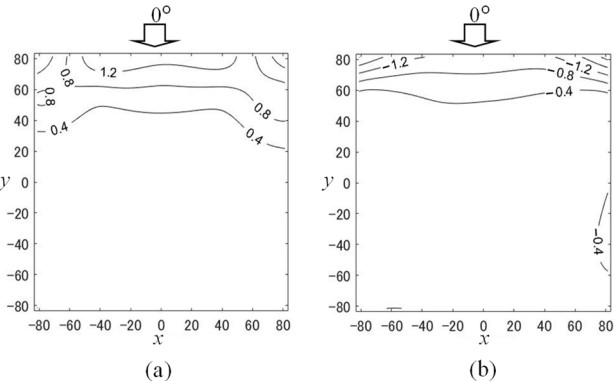

(a)  (b)

**Figure 12.** Contours of the maximum and minimum peak wind force coefficients at $\theta = 0°$: (**a**) Maximum; (**b**) minimum.

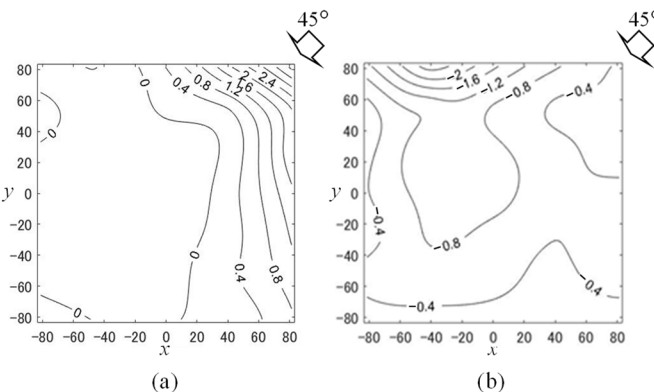

**Figure 13.** Contours of the maximum and minimum peak wind force coefficients at $\theta = 45°$: (**a**) Maximum; (**b**) minimum.

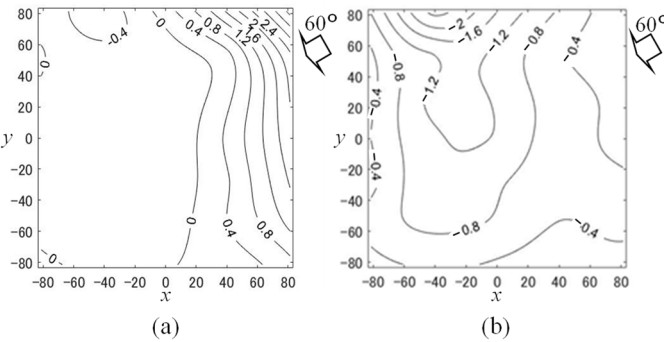

**Figure 14.** Contours of the maximum and minimum peak wind force coefficients at $\theta = 60°$: (**a**) Maximum; (**b**) minimum.

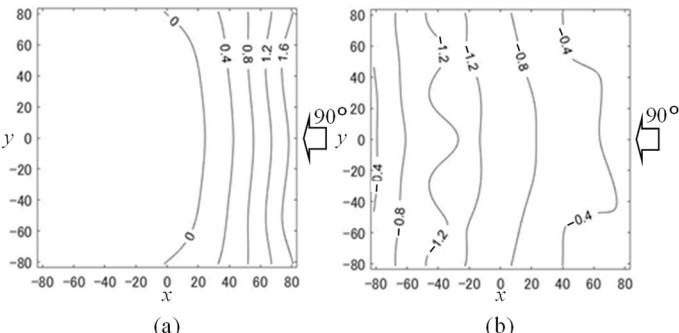

**Figure 15.** Contours of the maximum and minimum peak wind force coefficients at $\theta = 90°$: (**a**) Maximum; (**b**) minimum.

The contours of the most critical maximum and minimum peak wind force coefficients among all wind directions, $\hat{C}_{f,cr}$ and $\check{C}_{f,cr}$ are shown in Figure 16. In the figure, the results for only a quarter area of the roof are illustrated, considering the symmetry of the roof. It is clear that the values of $\hat{C}_{f,cr}$ are rather large near the corner and in strip areas along the eaves and verge. The magnitude of $\check{C}_{f,cr}$ is generally large along the verge. These features are similar to those observed for planar free roofs [5].

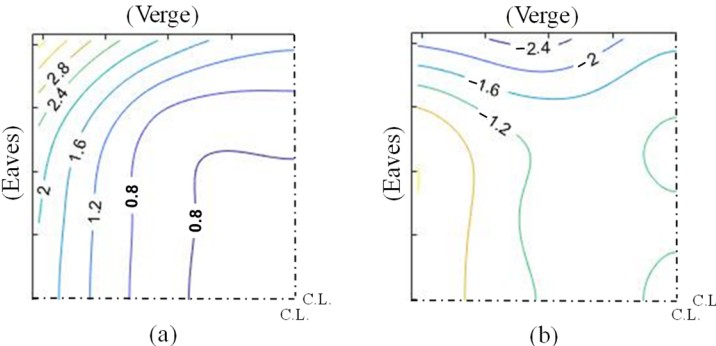

**Figure 16.** Contours of the most critical maximum and minimum peak wind force coefficients among all wind directions: (**a**) Maximum; (**b**) minimum.

Based on the results of Figure 16, we have proposed the positive and negative peak wind force coefficients, $C_{f\_pos}$ and $C_{f\_neg}$, for the design of cladding of the vaulted free roof under consideration. The results are shown in Figure 17. The roof is divided into several zones, $R_a$–$R_d$ for $C_{f\_pos}$ and $R_a$–$R_c$ for $C_{f\_neg}$, and the values of $C_{f\_pos}$ and $C_{f\_neg}$ are determined based on the maximum and minimum values of $\hat{C}_{f,cr}$ and $\check{C}_{f,cr}$ in each zone. The values correspond to a tributary area of $A_c = 1$ m$^2$, as mentioned above. The zoning is determined in reference to that for gable roofs [5] and that for cylindrical roof of enclosed buildings [8].

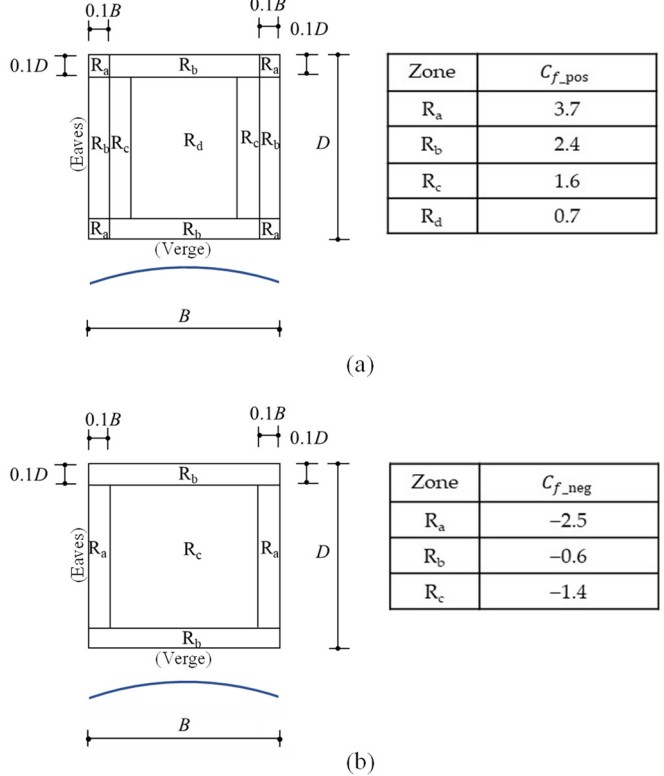

| Zone | $C_{f\_pos}$ |
|---|---|
| $R_a$ | 3.7 |
| $R_b$ | 2.4 |
| $R_c$ | 1.6 |
| $R_d$ | 0.7 |

(a)

| Zone | $C_{f\_neg}$ |
|---|---|
| $R_a$ | −2.5 |
| $R_b$ | −0.6 |
| $R_c$ | −1.4 |

(b)

**Figure 17.** Positive and negative peak wind force coefficients for designing cladding: (**a**) Positive peak wind force coefficient; (**b**) negative peak wind force coefficient.

## 6. Concluding Remarks

The design wind loads for a vaulted free roof with a square plan and a rise/span ratio of 0.1 have been discussed on the basis of a wind tunnel experiment, in which the wind pressure distributions on one half area of the roof were measured at various wind directions. It was assumed that the roof was supported by two rigid frames, each of which

consisted of two columns and a beam and carried the wind loads acting on the half area. When discussing the wind force coefficients for the main wind force resisting system, we focused on the bending moment $M$ at the windward column base and the axial force $N$ induced in the leeward column as the load effects.

The load effects became the maximum in a diagonal wind (i.e., $\theta = 60°$) not in a normal wind (i.e., $\theta = 90°$). This is because large positive wind forces acted on the windward corner area due to impinging winds and large suctions were induced along the verge due to a conical vortex. The distributions of equivalent static wind force coefficients that generated the maximum load effects were obtained by using the LRC method proposed by Kasperski [18]. Based on this distribution, we proposed a simple model of wind force coefficients for designing the main wind force resisting system in the framework of the conventional gust effect factor approach. That is, the roof was divided into three zones ($R_U$, $R_M$ and $R_L$), and constant values of wind force coefficients ($C_{f\_U}$, $C_{f\_M}$ and $C_{f\_L}$) were provided to these zones for the two load effects, $M$ and $N$. The design wind loads are provided by the product of the design velocity pressure, the proposed wind force coefficients and a gust effect factor. The gust effect factor can be obtained from an empirical formula (Equation (8)) which is a function of the turbulence intensity $I_{uH}$ at the mean roof height $H$ of the approaching flow.

The distributions of the maximum and minimum peak wind force coefficients, $\hat{C}_f$ and $\check{C}_f$, at all pressure taps were also obtained for various wind directions. The values of $\hat{C}_f$ and $\left|\check{C}_f\right|$ were large near the windward eaves and verge in diagonal winds. In particular, the values of $\hat{C}_f$ were very large near the windward corner. This is because large positive pressures acted on the roof's top surface due to impinging winds and large suctions acted on the roof's bottom surface due to flow separation. Based on the distributions of the most critical maximum and minimum peak wind force coefficients among all wind directions ($\hat{C}_{f,\mathrm{cr}}$ and $\check{C}_{f,\mathrm{cr}}$), we proposed the positive and negative peak wind force coefficients, $C_{f\_pos}$ and $C_{f\_neg}$, for designing the cladding of the free roof. Indeed, the roof was divided into several zones, and the values of $C_{f\_pos}$ and $C_{f\_neg}$ were provided to each zone.

This study focused on a case of $f/B = 0.1$, because this value of $f/B$ is often used in practical design according to our review of existing vaulted free roofs in Japan. However, larger $f/B$ values, such as $f/B \approx 0.2$, are also used in practice. Because the $f/B$ ratio affects the flow around the roof and the resultant wind pressure distributions significantly, similar discussion of design wind force coefficients should be made for higher $f/B$ ratios. This is the subject left for our future study.

**Author Contributions:** Conceptualization, Y.U. and W.D.; methodology, Y.U.; software, W.D.; validation, Y.U. and W.D.; formal analysis, W.D.; investigation, W.D.; resources, Y.U.; data curation, W.D.; writing—original draft preparation, W.D.; writing—review and editing, Y.U.; visualization, W.D.; supervision, Y.U.; project administration, Y.U.; funding acquisition, W.D. All authors have read and agreed to the published version of the manuscript.

**Funding:** This research was funded by Nohmura Foundation for Membrane Structure's Technology (Research Grant of FY 2021).

**Acknowledgments:** The experimental data used in this study were supplied by Japan Exterior Industrial Association. The wind tunnel experiments were carried out by Messrs. Tetsuro Yambe and Seiya Gunji, former graduate students of Tohoku University.

**Conflicts of Interest:** The authors declare no conflict of interest. The funders had no role in the design of the study; in the collection, analyses, or interpretation of data; in the writing of the manuscript, or in the decision to publish the results.

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
