# Peer review of "Discussion of Design Wind Loads on a Vaulted Free Roof"

_2674-032X, doi:10.3390/wind2030026_

Round 1

Reviewer 1 Report

This article discusses the design wind force on a Vaulted Free Roof through the wind tunnel tests. It is an interesting topic for the wind-engineering researchers and the topic fits in the scope of this journal. Following are some comments or suggestions for the authors to response.

 Comment 1: In Fig.4, the power spectrum density of wind velocity is not shown ?

Comment 2: It is advised to quantitatively compare the wind tunnel test with the CFD results in Fig.8 and Fig. 9.

Comment 3: Line 300, the definition of maximum peak values of ?∗(?) and ?∗(?) is unclear, how did the author obtain peak value ?∗(?) based on the time history of wind force coefficients?

Comment 4: The value of contour line and tick label number of x and y-axis in Fig. 13-16 is too small, and image resolution of these contours is low.

Comment 5: Reynolds number would significantly affects the flow structure near the boundaries. Did the authors consider Reynolds number effect for the practical design wind force

Comment 6: For the extreme wind force coefficient, please consider the method proposed by Hui et al. 2017.

Hui, Y., Tamura, Y., and Yang, Q.S., 2017. Estimation of extreme wind load on structures and claddings. J. Eng. Mech.(ASCE) 143(9), 0401708.

Reviewer 2 Report

The paper contains a detailed experimental program of obtaining wind load data for vaulted roofs. I find the paper of high significance for researches in the field of numerical analysis, where detailed and trusted experimental data is crucial. The paper is well written. 

Reviewer 3 Report

It is a nice piece of work that will be of good interest to fluiddynamics and architecture community.

1) In Fig. 2b, both the filled arrows indicate theta = 0degr. I suppose one of them should be 90 degr.

2) In line 162, authors state that "It is expected that the columns do not affect the flow around the roof and the resultant pressure distribution on the roof significantly because the width is only 7 mm.". However, I suppose the flow disturbance generated by two 7mm-wide rectangular columns is not negligible for 166mm-by-166mm roof because experiments were performed in subsonic wind tunnel where pressure disturbance from the columns propagates in any direction. Even though the extra columns were small, resulting pressure disturbance can be significant. So, authors should provide any evidence, or estimation of pressure disturbance magnitude, that the pressure disturbance from these columns are negligible.

Another concern is that these extra columns were positioned at Area 2 (-y area), asymmetric to x-axis judging from Fig. 3a. Authors state that Area 2 pressure distribution was obtained assuming "the symmetry of the model." (line 210). In case of theta = 0 degr measurement, Area 2 pressure was obtained by switching theta to 180 degr, substituting Area 1 data(theta=180) as Area 2 pressure(q=0). In this case, the Area 1 bottom-side pressure in theta= 180 case is apparently within the wake flow of extra columns and, I suppose, subsequent wake pressure disturbance is visible in Fig. 5b around (x,y)=(80,-20 to -40).

Reviewer 4 Report

This paper investigates the wind loads on a vaulted free roof through wind tunnel experiments. The paper is in general well organized and written. The results are interpreted and discussed in detail. The information can be useful for the wind-resistant design of similar vaulted free roofs.

1. The abstract is lengthy. However, the motivation and main innovation of this paper are not very clear from the abstract.

2. The organization of the introduction can be improved. For example, the introduction should start with the background while the first paragraph can be moved to the end of the literature review.

3. It is not clear why are some words highlighted in red color.
